# A Systematic Review of Factors That Influence Parents’ Views and Practices around Routine Childhood Vaccination in Africa: A Qualitative Evidence Synthesis

**DOI:** 10.3390/vaccines11030563

**Published:** 2023-03-01

**Authors:** Edison J. Mavundza, Sara Cooper, Charles S. Wiysonge

**Affiliations:** 1Cochrane South Africa, South African Medical Research Council, Cape Town 7500, South Africa; 2School of Public Health and Family Medicine, University of Cape Town, Cape Town 7925, South Africa; 3Department of Global Health, Stellenbosch University, Cape Town 7505, South Africa; 4HIV and Other Infectious Diseases Research Unit, South African Medical Research Council, Durban 4091, South Africa

**Keywords:** childhood vaccination, parents, caregivers, Africa, views and practices

## Abstract

A Cochrane review which explored the factors that influence caregivers’ views and practices around routine childhood vaccines worldwide was conducted by Cooper and colleagues. After sampling 154 studies that met their inclusion criteria, the authors included 27 studies in their synthesis, of which 6 were from Africa. The aim of the current review was to synthesise all 27 studies conducted in Africa. We wanted to determine if the inclusion of additional African studies will change any of the themes, concepts or theory generated in the Cochrane review. Our review found that parents’ views and practices regarding childhood vaccination in Africa were influenced by various factors, which we categorised into five themes, namely, ideas and practices surrounding health and illness (Theme 1); social communities and networks (Theme 2); political events, relations, and processes (Theme 3); lack of information or knowledge (Theme 4); and access-supply-demand interactions (Theme 5). All of the themes identified in our review were also identified in the Cochrane review except for one theme, which was lack of information or knowledge. This finding will help to promote vaccine acceptance and uptake in Africa by developing and implementing interventions tailored to address lack of knowledge and information around vaccines.

## 1. Introduction

It is estimated that more than 5.2 million children younger than the age of five around the world die every year [1]. It is also estimated that 1.5 million of these children die from vaccine-preventable diseases [2]. Most of these deaths take place in low and middle-income countries (LMICs) [3]. Vaccination is one of the most effective public health interventions used for controlling and eliminating life-threatening infectious diseases [1]. Vaccination programmes are responsible for the worldwide eradication of smallpox and the significant reduction in disability and death from diseases such as diphtheria, measles, polio, rubella, and tetanus [4,5,6]. In addition, vaccination also provides a range of other benefits, such as prevention of drug resistance and contribution to overall economical and societal well-being [1]. Vaccination is thus regarded as one of the greatest public health achievements of the 20th century [4,7,8,9,10]. For the vaccination programme to be successful, a high level of vaccination uptake is required, and if a sufficient proportion of a population is vaccinated, protection is also provided to the unvaccinated individuals through herd immunity [5,11]. 

The World Health Organisation (WHO) recommends at least 10 vaccines for routine immunisation for children [12]. Annually, more than 100 million children are vaccinated against diseases such as diphtheria, tetanus, pertussis, tuberculosis, polio, measles, and hepatitis B [10,13]. However, over 19 million children per year do not receive all of the recommended vaccines [13], contributing to numerous vaccine-preventable disease outbreaks and child deaths [5,6,7]. LMICs bear the largest proportion of under-vaccinated or non-vaccinated children [14]. Major factors that contribute to lower childhood vaccination coverage include lack of access to vaccination services, missed opportunities for vaccination during healthcare visits, and vaccine hesitancy [15,16]. Vaccine hesitancy is considered an important driver contributing to low levels of vaccination coverage in many settings [7,17]. Vaccine hesitancy is defined as a motivational state of being conflicted about or opposed to vaccination [16]. It is influenced by factors such as complacency (the person does not see a need and value for the vaccine), confidence (individual’s confidence in the vaccine), and convenience (access to vaccines) [18,19,20]. In 2019, the WHO identified vaccine hesitancy as one of the top 10 threats to global health [21]. 

While there is increasing attention being paid to the demand side of vaccination, we currently have limited understanding of the factors that influences caregivers’ views and practices regarding routine immunization among children in Africa [22]. Qualitative research can contribute to the understanding of these factors, as it is a well-suited tool to study people’s beliefs, behaviours, and decision making. It can also help inform policy and practice, including the development of tailored, effective, and acceptable interventions that can promote acceptance and uptake of childhood vaccination. Cooper and colleagues conducted a Cochrane qualitative systematic review exploring the factors that influence the views and practices of the caregivers regarding routine childhood vaccines worldwide [5]. Due to the large number (*n* = 145) of studies that met their inclusion criteria, the authors used pre-specified sampling method to sample studies that they included in their synthesis. They orderly sampled studies based on 3 criteria: ‘conceptual richness’, ‘relevance’, and ‘geographical spread’ of the included studies. After the sampling process, only 6 studies conducted in Africa (out of the total 27 studies conducted in Africa that met the inclusion criteria) were included in their synthesis. In this current review, we therefore synthesise the findings from all 27 studies conducted in Africa. We were interested in whether the inclusion of additional studies conducted in Africa changed any of the themes, concepts or theory generated in the larger Cochrane review. Specifically, our aim in this review was to explore the factors that influence the views and practices of the caregivers on routine childhood vaccines in Africa, and to compare and contrast our findings with those of the Cochrane review. We anticipated that there would findings that potentially strengthen, but at the same also differ or nuance, the findings from the larger Cochrane review. Ultimately, this current review will provide insights into the potential generalisability of the findings generated in the larger Cochrane review, and in turn facilitate a more comprehensive understanding of the factors that influence parents’ views and practices around routine childhood vaccines in Africa. The continent currently has one of the highest burdens of vaccine preventable diseases and lowest childhood vaccination coverage. The focus of this review is therefore important and timely. 

## 2. Materials and Methods

The methods of the larger Cochrane review by Cooper and colleagues are reported elsewhere [5]. In summary, the review included qualitative studies of different designs such as case studies, ethnography, grounded theory studies, phenomenology, and qualitative process evaluations. The included studies used qualitative methods for both data collection and data analyses. The authors included studies conducted among informal caregivers or parents. They defined an ‘informal caregiver’ as a person who was directly involved in care of the child or who was responsible for the decision to vaccinate or not to vaccinate the child, or who had a responsibility to take the child for vaccination. The Cochrane review included studies that examined parents’ or informal caregivers’ views, experiences, acceptance, or hesitancy about routine childhood vaccination. The authors included studies conducted in any setting around the world where childhood vaccination is offered. They searched MEDLINE (Ovid, MI, USA), Embase (Ovid, MI, USA) CINAHL (EBSCO, Ipswich, MA, USA), Anthropology Plus (EBSCOI, pswich, MA, USA), Web of Science Core Collection (Clarivate Analytics), PsycINFO (Ovid, MI, USA) databases for eligible studies from 1974 to July 2020, together with reference checking and citation searching to identify additional studies. They found 145 studies, described in 176 publications (173 articles and 3 books), that met their inclusion criteria. Of these studies, 27 were conducted in Africa and 6 were sampled to be included in their synthesis. 

In the current review we focus on the 27 publications conducted in Africa. Two authors (EM and SC) independently extracted data from all 27 African studies using a structured and standardised data extraction form. Extracted data included study author, country of study, type of participants, type of vaccine, methodology used, and views and practices.

For the analysis we used a ‘best fit’ framework synthesis approach [23,24]. This method provides a way to test, reinforce and build on an existing published model. It involves identifying a framework of a priori themes and coding data against that framework. Data that does not ‘fit’ or cannot be accommodated within the framework is analysed thematically to generate new themes and/or potentially adapt the original a priori framework. The larger Cochrane review had developed a four-theme framework for understanding the factors influencing caregiver views and practices regarding childhood vaccination. We used this as our a priori framework and analysed the data against it. The best fit” framework synthesis approach allowed us to test this framework and explore how it might be potentially strengthened and/or refined by data from a larger number of studies conducted in Africa.

## 3. Results

### 3.1. Results of the Search 

A total of 27 studies conducted in Africa were identified from the list of the studies that met the inclusion criteria in the Cochrane review by Cooper and colleagues [5].

### 3.2. Description of the Studies

The characteristics of the included studies are summarized in Appendix A, Table A1. 

#### 3.2.1. Setting

All studies were conducted exclusively on the African continent, except for two studies which were multi-continental [25,26]. Those conducted exclusively in Africa were conducted in: Burkina Faso (*n* = 2) [27,28], Chad (*n* = 1) [29], Cameroon (*n* = 1) [30], Ethiopia (*n* = 6) [31,32,33,34,35,36], Gabon (*n* = 1) [37], Guinea (*n* = 3) [38,39,40], Gambia (*n* = 1) [41], Malawi (*n* = 1) [42], Nigeria (*n* = 3) [43,44,45], South Africa (*n* = 2) [46,47], Togo (*n* = 1) [48], Uganda (*n* = 2) [49,50], and Burkina Faso and Central African republic (*n* = 1) [51]. The remaining two studies were multicontinental: Angola, Ethiopia, India, Nepal, Nigeria, Pakistan, and Rwanda (*n* = 1) [25], and Botswana, Dominican Republic, and Greece (*n* = 1) [26].

The included studies focused on routine immunisation programmes (*n* = 18) [26,27,28,29,31,33,34,35,36,37,38,40,41,46,47,48,49,50]; specific vaccines such as polio vaccine (*n* = 4) [25,32,43,45], measles vaccine (*n* = 2) [39,44], tetanus vaccine (*n* = 1) [42], and hepatitis B vaccine (*n* = 1) [51]; and a combination of polio mass vaccination campaigns and routine immunisation (*n* = 1) [30]. 

#### 3.2.2. Respondents 

All included studies were conducted among caregivers, defined as parents or others with primary responsibility for making vaccination decisions for children. The studies were conducted among mothers only (*n* = 11) [28,29,31,34,35,36,37,38,41,44,48]; mothers and fathers (*n* = 8) [25,30,32,40,43,45,49,51]; mothers, fathers, grandmothers, and grandfathers (*n* = 1) [27]; mothers, grandmothers, aunts, and sisters (*n* = 1) [47]; and pregnant women (*n* = 1) [42]. The remaining five studies did not specify the type of caregivers they collected data from [26,33,39,46,50]. 

#### 3.2.3. Methodology 

The studies conducted in-depth interviews (*n* = 2) [25,32]; focus group discussions (*n* = 6) [33,36,39,40,48,50]; in-depth interviews and focus group discussions (*n* = 8) [26,27,28,29,42,44,47,49]; in-depth interviews and observations (*n* = 6) [31,34,37,38,43,46]; in-depth interviews, focus group discussions, and observations (*n* = 3) [35,41,51]; and in-depth interviews, observations, and informal discussions (*n* = 2) [30,45]. 

### 3.3. Review Findings 

Five themes were identified through our analysis: ideas and practices surrounding child health and illness (Theme 1); social communities and networks (Theme 2); political events, relations and processes (Theme 3); lack of information or knowledge (Theme 4); and access-supply-demand interactions (Theme 5). Table A2 shows the summary of the review findings. 

#### 3.3.1. Theme 1: Ideas and practices surrounding child health and illness

##### Religious Beliefs 

Four studies that were conducted in Ethiopia [35], Malawi [42], Nigeria [45], and Togo [48] found that religious beliefs that some parents had led them to less accept childhood vaccination. Some parents in Ethiopia believed that diseases were caused by God and will be prevented by God, 


*“it is only God who is able to bring all kinds of illness to people”*
(Ethiopia, participant quote) [35]

*“who are we to prevent illness, that is God’s task”*. (Ethiopia, participant quote) [35]

Similar beliefs were mentioned by some parents from Togo and Nigeria: 

*“It is God, you can’t prevent what he gives you”*;(Togo, participant quote) [48]

*“If God wishes, the one who had immunisation will be sick but the one who didn’t have immunisation will be in good health”*. (Nigeria, participant quote) [45]

##### Beliefs about the Benefits and Risks of Vaccines

Parent’s ideas and practices regarding routine childhood vaccination may be influenced by their beliefs regarding the benefits of vaccination and their risk perception towards vaccines. Various studies revealed that many parents accepted childhood vaccination because they believed that it was beneficial for the health and wellbeing of their children [27,28,34,36,37,40,42,46,48,50,51]. A study in South Africa found that some parents were more accepting of childhood vaccination because they wanted to protect their children from illness; 

*“I only do it for the child’s sake because I know that he will be safe from getting sick”*. (South Africa, participant quote) [46]

Another study conducted in Malawi among pregnant women found that they accepted maternal vaccination against influenza because they were protecting themselves and their unborn children [42]. In addition to illness, prevention of deaths was mentioned as a reason some parents were accepting vaccination for their children in Ethiopia; 

*“Previously most of our children died of measles and tetanus, but after using vaccination services those illnesses and deaths were prevented”*. (Ethiopia, participant quote) [36]

Studies conducted in Ethiopia [36], Uganda [50], and Burkina Faso [28] reported that some parents accepted vaccination for their children because it was preventing diseases and promoting the development of their children. A study in Burkina Faso found that many parents accepted childhood vaccination because it eradicated diseases that were affecting them before; 

*“Vaccination is good because through vaccination all diseases that existed when we were little are gone today–for example measles and smallpox which caused epidemics when we were small and in school”*. (Burkina Faso, participant quote) [28]

However, a study in Nigeria reported that some parents were hesitant towards childhood vaccination because they believed that it was unnecessary or ineffective; 

*“There is no difference seen between those children that are immunised and those that are not immunised”*. (Nigeria, participant quote) [44]

Several studies found that many parents defaulted on their children’s vaccination because they were concerned about the occurrence of the side effects after vaccination [27,28,29,33,35,36,37,39,40,42,43,44,46,47,48,51]. Some parents in Burkina Faso [27,28] and South Africa [46] mentioned a fear of the pain their children were experiencing after receiving injections as the reason they discontinued with vaccination: 

*“Some are afraid of children crying after vaccination, so they no longer return to vaccination”*; (Burkina Faso, participant quote) [28]

*“The infant is too young and it felt like they are in deep pain when they were injected”*. (South African participant quote) [46]

A study conducted in Malawi revealed that several pregnant women also cited a fear of pain that is associated with receiving injections as a reason they avoided maternal vaccination [42]. 

Several studies found that some parents discontinued vaccination for their children because of the occurrence of sickness, soreness, formation of an abscess around the site of injection, and other reactions they have witnessed after vaccinating their children [33,35,36,39,48,51]. For example, some parents in Ethiopia mentioned that sickness was the reason they defaulted on their children’s vaccination: 

*“My child was seriously sick when she took the vaccination. It was hard to continue that way for the next schedule since the illness was very serious leading her to a serious leg spasm and high fever, hence my husband and I discussed that and decided to quit it”*; (Ethiopia, participant quote) [33]

*“After the third immunisation shots on the thigh of the child, she became febrile and swollen at the site of immunisation; for this reason, my husband was angry at me since I took a well child and brought her back sick following vaccination. Therefore I did not bring the child for the last immunisation”*. (Ethiopia, participant quote) [36]

One study in Guinea [40] and the other one in Ethiopia [35] reported that the occurrence of abscess at the site of injection was cited by some parents as a reason for discontinuation of vaccination of their children. Another study in Nigeria revealed that some parents refused to vaccinate their children because of boils which appeared around the site of vaccination [43]. Other side effects that were reported by some parents as reasons for non-vaccination of their children were occurrence of fever, diarrhoea, inflammation of the throat, and swellings around the injection site [29,37]. Studies in Nigeria [43,44] and Burkina-Faso [28] found that some parents declined to vaccinate their children because they believed that vaccines would lead to infertility.

However, other studies conducted in Gambia [41], Ethiopia, [31], Gabon [37], Uganda [50], and Burkina Faso [51] found that some parents were more accepting of childhood vaccination despite the occurrence of side effects after vaccination. For example, some parents in Uganda continued to vaccinate their children despite their concern about side effects because it was a very important thing to do: 

*“Some of us know that immunisation is very important, so concerns have not stopped us from immunising our children. There is a reduction of measles in our area, so I follow what the trained HCWs tell me”*. (Uganda, participant quote) [50]

These parents perceived side effects as a sign of the effectiveness of the vaccination. Some parents in Ethiopia interpreted the pain of vaccination as a proof of the worth of the vaccine, and the occurrence of diarrhoea was perceived as the discomfiture and defeat of the disease in a bodily form [31]. 

#### 3.3.2. Theme 2: Social Communities and Networks

Several studies across Africa found that vaccination views and practices of the social networks where caregivers resides shaped their views and practices around routine childhood vaccination [27,35,41,42,43,48,49,51]. Three studies revealed that caregivers’ acceptance of childhood vaccination was influenced by community members, including other parents, relatives, peers, neighbours, and other important members of the community [35,43,49]. A study in Ethiopia reported that some parents vaccinated their children after seeing their neighbours and friends doing it [35]. Another study in Uganda revealed that some mothers were influenced by their spouses to vaccinate their children, 

*“As for me, I make sure that when my wife is pregnant she attends the antenatal clinic as required and is also immunised because she usually tells me when she is immunised. Also after she gives birth I make sure she takes the children for immunisation on the dates written on the immunisation card”*. (Uganda, participant quote) [49]

Some parents in Nigeria were influenced by other parents to vaccinate their children: 

*“I have seen parents taking their children for vaccination so it makes me accept it”*. (Nigeria, participant quote) [43]

A study in Burkina Faso found revealed that some parents were more accepting of childhood vaccination because they took it as a social practice, it was practised by the whole society; 

*“When they tell you to come out [to vaccinate your child] one day and everybody comes out… you can’t refuse!”*. (Burkina Faso, participant quote) [51]

However, several studies reported that some parents did not vaccinate their children because they were influenced by their family members and important figures of their communities [27,43,48,49]. In Nigeria, some parents decline childhood vaccination because the teachers influenced them, 

*“…In Islamiyya school, they [teachers] would tell people not to give their children the vaccine because there is something bad in it. It is because of this, that here in Locus [her neighborhood], in one in every five houses, you will find people who refuse to immunise their children”*. (Nigeria, participant quote) [43]

Other parents in the same country refused routine childhood vaccination because significant people of the community did not endorse it: 

*“I never heard the Sultan [of Sokoto] explain this polio vaccine on the radio or anything. When they came to my house, I told them I don’t want them to do it for my children and the health worker, she just went out…”*. (Nigeria, participant quote) [43]

In Malawi, some parents declined to vaccinate their children because they were influenced by the members of their churches who never take any medication to treat any diseases; 

*‘‘…[Members of my church] don’t take medication. So for all my ten children, I have not gone to receive this vaccine [tetanus toxoid vaccine]. Neither the children nor I have received the vaccine”*. (Malawi, participant quote) [42]

Two studies found that some mothers failed to vaccinate their children because they were stopped by their husbands [48,49]. In Uganda, some mothers were stopped from vaccinating their children because the husband had more faith in traditional medicine than in immunisation. 

*“My wife is pregnant but she has not been immunised. She has a four-year old child and she talks about immunising the child but I stop her from doing it. For me I don’t believe in it. As you can see, I am a mature person but I did not grow up because of that (immunisation). It was better for me to use traditional medicine to treat fever for example, but because these days the fever is very strong, I now use tablets (for treatment). Even these injections (from immunisation) paralyse people I know, and we also see them in books and in pictures”*. (Uganda, participant quote) [49]

Some mothers in Togo failed to take their children for vaccination because their husbands were in possession of the immunisation booklets, and they did not give them a permission to go to the vaccination centres; 

*“Husbands keep the cards … Some husbands don’t want their wives to go”*. (Togo, participant quote) [48]

A study in Uganda found that some husbands who refused their wives from taking the children for vaccination were influenced by their older relatives, 

*“Like I explained before about some elderly women who claim children will become lame after immunisation, some men use that excuse because they had ever heard of it while still young. So when they grow up and get children they say the children will become lame or get brain damage. That is why you see some children when they get measles they almost die because the husband refused the wife to take children for immunisation”*. (Uganda, participant quote) [49]

However, some mothers in Burkina Faso did not take their children for vaccination despite the fathers’ permission to vaccinate the children; 

*“It is the father [who accepts], it is the mothers who refuse and some women leave and don’t get their child vaccinated”*. (Burkina Faso, participant quote) [27]

#### 3.3.3. Theme 3: Political Events, Relations, and Processes

##### Generalised Decline in Trust of Authority and Expert Systems

One study conducted in Nigeria found that the low level of trust that some parents had in the institutions or systems involved in vaccination is due to a more general decline in trust in authorities and expert systems [43]. Some parents in this study were vaccine hesitant due to their distrust of government [43]. This study revealed that some parents did not vaccinate their children against polio because of their distrust of the Federal Government under President Olusegun Obasanjo, and the heads of both the Ministry of Health and the National Programme on Immunisation, who were held responsible for the deteriorating health conditions in northern Nigeria [43]. 

##### Agendas and Interests Underpinning the Expert Systems Implicated with Vaccination

A total of four studies found that socio-political agendas or interests related to vaccination programmes were concern for some parents [29,43,50,51]. Some parents in Burkina-Faso [51] and Chad [29] were particularly concerned with what they believed was financial interests which was the motive behind vaccination programmes, which in turn led them to question the motivations of those conducting or promoting vaccination. Some parents in Chad believed that childhood vaccinations were conducted for the involved workers to gain finances:

*“…..this vaccination business is a way to get work for some, to make money for others and all this at the expense of the health of our children” (Chad, participant quote) [29]; “For me, polio is an organised business from the high hierarchy to the last vaccinators who administers the drops. Everyone finds his interest and that’s it. This practice makes me doubt about the efficacy of poliomyelitis vaccine”*. (Chad, participant quote) [29]

In Burkina Faso, some parents were less accepting of childhood vaccination because they believed that it was a money-making business for the governments and pharmaceutical companies: 

*“Vaccination made children sick, compelling parents to seek medical care at these centers, which collected consultation fees and channeled them into state coffers” (Burkina Faso, participant quote) [51]; “Europeans don’t want to throw [vaccines]… out, they come to vaccinate us with them. These medicines make everyone sick. And when you fall sick, then you will turn around and buy more of their medicines”*. (Burkina Faso, participant quote) [51]

Some parents in Nigeria [43] and Uganda [50] believed that there were socio-political agendas behind childhood vaccination programmes. These parents believed that vaccination programmes were not meant to prevent diseases but were intended to reduce their population or to kill them: 

*“No, I don’t allow the people to do polio vaccination for my children in the house because there is a problem in it, such as that European people want us to reduce our numbers, to stop us from giving birth”* (Nigeria, participant quote) [43]; *“When the temperatures become high, we suspect that either the vaccine was expired or there was no vaccine in the injection. We think whites want to kill us”*.  (Uganda, participant quote) [50]

##### Current and Past Controversies

Studies in Nigeria [43,44,45], Guinea [39], and Uganda [50] found that many parents were hesitant to vaccinate their children due to the current and past controversies around vaccines or any health-related issues more broadly. Some parents were concerned about the adverse effects they previously experienced following vaccination of children: 

*“There was mass immunisation in 1990s, it killed children. Children died one after another and people said they were being killed by immunisation” (Uganda, participant quote) [50];“Look at what happened in Kano. They were testing the CSM [cerebrospinal meningitis] vaccine and killed so many children. Therefore, some people believe it is not safe”*.  (Nigeria, participant quote) [44]

A study in Guinea reported that some parents were reluctant to vaccinate their children after the outbreak of Ebola in West Africa between 2014 and 2016, which led to an associated fear of attending health structures such as hospitals [39]. Two controversies in Nigeria were identified by some parents as some of the reasons they were hesitant towards childhood vaccination [43,45]. The first controversy that took place in Kano State in 1996 during the epidemic of meningitis in the country was the Trovan controversy, which involved Pfizer in a clinical trial of the antibiotic Trovan. After the deaths of several children during this trial, a lawsuit was filed against Pfizer for failing to fully inform participants of the associated risks, for administering the control drug at a low dose, and for concealing the availability of the approved treatment for meningitis [45]. The second controversy, which took place between 2003 and 2004, was around the boycott of polio vaccine in Nigeria. The boycott took place in five Northern Nigerian states after various political and religious leaders raised concerns about the safety of the oral polio [43]. 

##### Marginalisation, Inadequate Public Services, and Priority Misalignment

Four studies, three in Nigeria [43,44,45] and one in Uganda [50], found that some parents mistrusted institutions implicated in vaccination and they were hesitant towards childhood vaccination because matters that they themselves prioritized were neglected. Some parents refused vaccination for their children because they were questioning the government’s motive for focusing only on childhood diseases: 

*“The government has a hidden agenda on immunisation. I think like this because there are so many other diseases that need assistance but they only talk about childhood immunisation”*. (Nigeria, participant quote) [44]

Other parents questioned the reasons behind vaccination services being offered free of charge while other healthcare services incurred costs: 

*“No, I don’t allow my children to have the vaccine because I don’t trust the vaccine. Because they said they are going to do it free of charge. And if we go to the hospital, we have to buy medicine and it is costly there. But this one is free of charge. In the hospital, your child can die or your brother can die if you don’t have money. My children have had measles vaccine, but this polio vaccine, I won’t allow it…If I believe in polio or go to the hospital and have medicine free of charge, like this polio, I can accept the polio vaccine. But if I have to pay for medicine in the hospital, I will not accept this one”*. (Nigeria, participant quote) [43]

In Nigeria, some parents declined polio vaccination which was not seen as a primary health problem for their children, but more prioritised and promoted than other healthcare services: 

*“And we are looking for medicine in the hospital to give to our children and we can’t get it but this one, they are following us to our houses to give it. I don’t trust this polio vaccine”*. (Nigeria, participant quote) [43]

#### 3.3.4. Theme 4: Lack of Information or Knowledge 

Parents’ ideas and practices regarding childhood vaccination in Africa may be influenced by the level of their knowledge around childhood vaccination. Several studies found that some parents were less accepting of childhood vaccination because they did not have enough information to understand the benefits of vaccination or to know the process of getting their children vaccinated [28,29,33,34,36,39,40]. For example, a study in Chad revealed that some parents were not informed about vaccination; they only heard about it from the ordinary people in the streets: 

*“Vaccination, I heard about it in street talks from people who do not know more than me. So nobody really told me what it is”*. (Chad, participant quote) [29]

Some parents in Ethiopia mentioned that during their visits to the vaccination centres they did not receive any information about advantages and disadvantages of vaccines; they were only informed about their next appointment date [33]. In Burkina Faso, some parents who did not receive information about the importance of the vaccines believed that they were wasting their time by taking their children to the vaccination centres [28]. A study that was conducted in Chad reported that some parents were complaining about their children being vaccinated but not told for which disease [29]. In another study, some parents indicated that if they were informed which diseases their children were vaccinated against, they would be willing to continue to vaccinate them: 

*“There is [a] need to differentiate the diseases for which we vaccinate our children, this will empower us to vaccinate, but if caregivers don’t know why to vaccinate, they won’t accept”*. (Guinea, participant quote) [39]

Some parents in Guinea also cited lack of knowledge, poor understanding of health education, and misperception of side effects of vaccines, as some of the reasons that discourage them from vaccinating their children fully [39]. 

In Ethiopia, some parents defaulted on their children’s vaccination because they did not know that they can vaccinate them through another approach after health workers did not show up at their houses: 

*“The health extension worker failed to show up for a home visit after giving the earlier doses at home”*. (Ethiopia, participant quote) [34]

A study conducted in Guinea reported that some parents missed vaccination for their children because they did not know that sick children could be vaccinated, or that a child needs vaccination five times by the age of one year [40]. Parents in many studies suggested that receiving enough information on the vaccines and the target diseases will improve childhood vaccination uptake [25,36,40,48,51]. 

#### 3.3.5. Theme 5: Access-Supply-Demand Interactions

##### Socio-Economic Challenges in Accessing Vaccination Services

Several studies found that some parents were vaccine hesitant due to socioeconomic challenges they were facing, including distance, transport money, household work, work/employment, and childcare constraints [27,28,31,32,33,35,37,44,47,49]. Some parents in Nigeria [44], South Africa [47], Burkina Faso [27], Ethiopia [33,35], Uganda [49], and Gabon [37] mentioned the long distance they had to travel, often by foot, to reach the healthcare facilities was the reason they defaulted on vaccination of their children: 

*“The health facility is very far away. That is why we only go once or twice”* (Nigeria, participant quote) [44]; *“the clinic is too far”*. (Ethiopia, participant quote) [35]

In Nigeria, some parents mentioned that the health facilities to vaccinate their children should be brought closer to them and they recommended the use of mobile clinics as the 

*“health facilities should be brought closer to the people”*. (Nigeria, participant quote) [44]

Studies in South Africa [47], Ethiopia [31], and Uganda [49] found that some parents failed to vaccinate their children due to the lack of transport fares to go for vaccination. For example, some parents in Ethiopia indicated that even though they wanted to vaccinate their children, they don’t have money to take their children to the vaccination centres, 

*“I didn’t go to health centre not because I lack interest to go, it is rather lack of money”*. (Ethiopia, participant quote) [31]

In Uganda, some parents indicated that they would rather use their money to buy food instead of paying for transport to a vaccination centre, 

*“If I don’t have food, how can I use Uganda shillings 2000 [approximately US$1] for a boda-boda [means of transport using motorcycle/bicycle] to go for immunisation?”*. (Uganda, Participant quote) [49]

Several studies revealed that many parents defaulted on their children’s vaccination because it was competing with other activities in the household or going to work [27,31,33,35,44,49,51]. A study in Ethiopia found that some parents discontinued with their children’s vaccination because they were busy with house chores during the vaccination period, 

*“my child discontinued due to a personal problem of being busy with my household chores and got no time to take my child to HC on schedules”*. (Ethiopia, participant quote) [33]

In the same country, Ethiopia, another study reported that some parents missed to vaccinate their children because they were busy with other social activities when the healthcare workers came to their villages [35]. In Nigeria, some fathers mentioned that their wives did not have time to take their children for vaccination because they were busy with farming and other household chores, 

*“Our women don’t have the time to stand in a queue and wait for their turn to immunise their children. They have to go to the farm and attend to other household activities”*. (Nigeria, participant quote) [44]

Three studies, two in Burkina Faso [27,51] and one in Uganda [49], found that some parents missed vaccination schedules because they were busy at work during the vaccination period. For example, some parents in Burkina Faso mentioned that although they wanted to take their children for vaccination, they were busy working in the market to make some money for their families, 

*“My children haven’t gotten all of their vaccinations…I have to work every day at the market, because my husband... works in our fields (50 km outside of Bangui). If I don’t go to the market, my family won’t have any money. I know that I had to bring my children to receive their vaccinations. But I couldn’t do it”*. (Central African Republic participant quote) [51]

Another study in Burkina Faso found that some parents who were willing to vaccinate their children were at work during vaccination and after work it is too late and vaccination is over [27]. 

Some parents in South Africa [47] and Ethiopia [32] missed vaccination of their children due to the lack of someone to take the children to the vaccination centre or to look after the other children or family members who cannot be left alone while they have gone to the vaccination centres. Some parents in Ethiopia mentioned that although they were interested in vaccinating their children, they defaulted because they did not have older child to look after other children when they go to the vaccination centres, 

*“I did not participate because of my own personal problem. I do not have an older child who cares for my kids”*. [32]

##### Undesirable Features of Vaccination Services and Delivery Logistics

Various studies found that parents were vaccine hesitant due to the undesirable features of vaccination services and delivery logistics [27,28,30,31,33,34,37,39,44,49,50]. Studies conducted in Burkina Faso [27], Cameroon [30], and Ethiopia [33,34] mentioned that some parents defaulted on their children’s vaccination because they had lost or forgotten vaccination booklets at home. For example, some parents in Ethiopia missed their children’s vaccination because they did not have the vaccination booklet with them when they visited vaccination centres, 

*“…..they never accept you without the vaccination card”* (Ethiopia, participant quote) [33]. Some parents in Burkina Faso have confirmed the requirements of the immunisation booklet in order to receive vaccination service,

*“Vaccination is not available to mothers who don’t have the child’s booklet. So they have to return home or go and get the booklet from the CSPS”*. (Burkina Faso, participant quote) [27]

Some parents in Nigeria failed to vaccinate their children because their births were not registered due to home birth, and they believed that a child without a birth certificate could not be vaccinated in government health facilities; 

*“We think that when we deliver at home the child cannot be immunised in the hospital”*. (Nigeria, participant quote) [44]

Many parents stated that they defaulted on their children’s immunisation because of the long waiting times they spend at the healthcare facilities before they can receive the service [28,30,31,34]. Some parents in Ethiopia failed to vaccinate their children despite going to the vaccination centres because they spent many hours in the centre but the health workers were busy with their personal matters and failed to vaccinate their kids even after they were done, 


*“I waited for five hours in the health centre until the health workers came from the funeral but even after that nobody paid attention to my child’s vaccination and me. So I came back without getting the service.”*
(Ethiopia, participant quote) [34]

In Burkina Faso, some parents complained that they come early to the vaccination centres, but the vaccination does not begin early, and they are made to wait for long hours before they can vaccinate their children. These parents also indicated that finally when the vaccination takes place often the order of arrival is not respected, parents who came late they receive service before them [28]. Some parents in Cameroon, mentioned that they come early in the vaccination centres but spend long hours to vaccinate their children because healthcare workers are very slow; 

*“A parent might come here very early in the morning just to weigh her child then you sit until twelve o’clock. As for me, I have already weighed my baby and now I am waiting to vaccinate because I want to go home as soon as possible. So they are not fast at all.”*. (Cameroon, participant quote) [30]

Lack of financial resources for the payment of hospital services, such as delivery costs, buying drugs, and vaccination booklets was mentioned by some parents as the reason they did not return to the hospital to vaccinate their children [27,37,39,44]. A study in Gabon found that some parents defaulted on their children’s vaccination because they owed hospital delivery costs and they were supposed to pay it before they can receive any further service; 

*“if you are already not able to pay the delivery, you cannot go back to the hospital; you are obliged to stay at home”*. (Gabon, participant quote) [37]

Some parents in Burkina Faso were concerned about the amount they should pay for the immunisation booklet before they can be able to vaccinate their children, 

*“It’s free [vaccination], there’s no problem, but you have to pay from 25 to 100 CFA francs for the vaccination booklet, and this is why lots of people don’t have their children vaccinated”*. (Burkina Faso, participant quote) [27]

In Nigeria, some fathers mentioned that healthcare workers collect money from their wives at the vaccination centres, and that is the reason why their children defaulted in their vaccination; because they did not have money to give these healthcare workers and suggested that government should make vaccination free [44]. In another study conducted in the Republic of Guinea, some parents indicated that they stopped vaccinating their children because they receive drug prescriptions after vaccination which they are unable pay; 

*“If you take your child to the hospital, saying that [she] had been vaccinated they will prescribe drugs that you cannot afford. Parents don’t have 100 Fr [20 cents US$ to pay for drugs”*. (Republic of Guinea, participant quote) [39]

Studies in Ethiopia [34], Gabon [37], Nigeria [44], and Uganda [49,50] revealed that some parents defaulted on their children’s immunisation due to the vaccine stockouts at healthcare centres. A study in Nigeria found that parents had to stop taking their children for vaccination because whenever they visited vaccination centres there were no vaccines available, 

*“We have to go three to four times. Each time we go, health workers tell us that they don’t have vaccines at the health facility. We have become tired of this, which is why we don’t bother going there anymore”*. (Nigeria, participant quote) [44]

In Ethiopia, some parents defaulted in their children’s vaccination because during their appointment dates there were no vaccines in the vaccination centres and they were given new dates, but they never returned; 

*“… on the third visit, I was told upon arrival that there was no vaccine in the health centre. The health worker gave me a new appointment date, but I haven’t gone there until now”* (Ethiopia, participant quote) [34]. *Some defaulting parents in Nigeria indicated that if vaccines were always available whenever they visited vaccination centres, they would be willing to go and vaccinate their children; “If the vaccines are always available, people can go at any time and receive the immunisation for their children”*. (Nigeria, participant quote) [44]

Two studies, one in Gabon [37] and one in Uganda [49], found that the fear of being bullied, mocked or stigmatised by other parents for their poor appearance or those of their children during their visits to the healthcare facilities was mentioned by some parents as the reason they defaulted on their children’s vaccination. Some parents in Gabon indicated that they were reluctant to take their children for vaccination because they were embarrassed/ashamed of their poor clothes that were not in good condition in front of other well-dressed parents; 

*“Sometimes you have a pair of shoes, that is not good. They already start to stare at you and immediately you feel embarrassed, …you feel ashamed if you see the others well dressed and yourself badly dressed, you are frustrated”*. (Gabon, participant quote) [37]

In Uganda some parents mentioned that they feared to take their children for vaccination because they were abused by healthcare workers for not carrying their children in clean and proper garments [49]. 

Two studies conducted in Uganda revealed that many parents were only willing to vaccinate their children if the vaccine was given through routine, but not mass, immunisation because they believed routine immunisation was safer [49,50]. 

##### Interactions with Frontline Healthcare Workers

Several studies found that parents were vaccine hesitant following experiences of mistreatment or poor communication from frontline healthcare workers. Some parents in Chad [29], Ethiopia [36], Republic of Guinea [39], and Uganda [50] complained about the state at which the healthcare workers were at and their attitude towards them during the vaccination process. For example, some parents in Ethiopia did not complete vaccination schedules of their children because whenever they visited vaccination centres they were poorly welcomed and argued with: 

*“The health professionals didn’t welcome the clients properly rather they argue with us and make an unacceptable dialogue. In order to avoid such types of arguments and unethical dialogue, we missed the rest of immunisation schedules. The HWs didn’t understand our problems”*. (Ethiopia, participant quote) [33]

In another study conducted in Chad with pastoralists, some parents complained about the filthy and drunk vaccination workers who are sent to their homes to vaccinate their children [29].

Three studies, two in Ethiopia [31,34] and one in Gabon [37] found that some parents were denied vaccination or mistreated because they have missed vaccination appointment or came late: 

*“The reason why we don’t take our children is the workers at the health centre don’t admit us if we have missed the date of the appointment. They insult us for being late. So we fear to go there for vaccination because they offended us”*. (Ethiopia, participant quote) [31]

Some parents in Gabon have confirmed the insults parents are subjected to for missing the appointment date: 

*“Next time they will insult you. They will ask you “Madame, where have you been?” They will insult you”*. (Gabon, participant quote) [37]

Another study conducted in Ethiopia confirmed that some parents did not bother to take their children for vaccination because they knew that they were not going to receive it after they have missed the appointment date [34]. 

Three studies found that some parents were mistreated or denied vaccination by the healthcare workers for failing to produce vaccination booklet [27,33,34]. Some parents in Ethiopia indicated that they discontinued to vaccinate their children after they were denied vaccination and humiliated for not being able to produce the immunisation booklet: 

*“the HWs let you down, they did not respect you! They use to say why you don’t keep the card just like your child! But, sometimes we lost the card! That is not deliberate! But they never accept you without the vaccination card. So, when one observes while health workers disgrace a mother who does not have a card, she will never go for vaccination if she lost the card too because she feels she will experience the same thing. So, people say “why do I receive such embarrassment while my child is healthy!” As a result, they prefer to discontinue the vaccination programme”*. (Ethiopia, participant quote) [33]

In another study conducted in Ethiopia, some mothers also mentioned that they were afraid to go to the vaccination centres because healthcare workers were going to humiliate them for losing the vaccination cards [34]. Some mothers in Burkina Faso confirmed that vaccination services were not available to the mothers who didn’t have their children’s vaccination booklets [27]. 

A study in Gabon found that some mothers defaulted on their children’s vaccination because they were mistreated by the healthcare workers over the condition of their children during their visits to the healthcare facilities for vaccination: 

*“There are women, who have babies, who suffer from malnutrition, who feel ashamed to bring theirs, because of this malnutrition, because if you arrive at the MCC and the baby is not in a good condition, they [MCC nurses] inevitably confront you because of the baby. That makes you feel ashamed”*. (Gabon, participant quote) [37]

Therefore, some of the parents in the same study indicated that they prefer to stay at home instead of taking their children for vaccination when they see that they are not thriving [37]. 

Some parents in Chad [29], Cameroon [30], and Ethiopia [33,35] mentioned the lack of or minimal communication of vaccination information by the healthcare workers as barriers to vaccination. In Chad and Cameroon, parents were concerned that the healthcare workers were just vaccinating their children without explaining any information: 

*“What we deplore is that when the vaccinators come here, they do not explain what they came to do. They only call the children and put the drops in their mouths. They are always in a hurry”* (Chad, participant quote) [29]; *“At this hospital, they are not really informed. They didn’t really give any detailed information here about vaccines”*. (Cameroon, participant quote) [30]

Similarly, parents in Ethiopia indicated that they were only informed about their next appointment date; 

*“The health providers administered injections to our children and told us only the next immunization day. No further information was given to us”*. (Ethiopia, participant quote) [33]

Three studies found that some parents defaulted on their children’s vaccination due to the poor communication skills of the healthcare workers that vaccinated their children [28,29,39]. Some parents in Guinea complained that vaccination workers were unable to explain which diseases they were vaccinating their children against: 

*“It is needed that they [health professionals] are able to explain the diseases which they vaccinate for, which would help mothers decide to vaccinate their children. But if they do not know why they vaccinate, caregivers will not want to accept the vaccine”*. (Guinea, participant quote) [39]

A study conducted among pastoralists in Chad found that some parents complained that they were sent dirty and untrained young people who knew nothing about health, instead of trained healthcare workers to come into their homes and vaccinate their children [29]. In Burkina Faso, some parents indicated that they were not happy with the healthcare workers that had vaccinated their children because they caused a lot of harm on their children by pushing the needle too deep [28]. These parents suggested that trainees should not be allowed to vaccinate children; they must first observe how nurses do it: 

*“It should be avoided to give children to the trainees who have not yet experience. Let them observe first how nurses do it before giving them children to be vaccinated”*. (Burkina Faso, participant quote) [28]

However, studies in Botswana [26], Cameroon [30], Chad [29], Ethiopia [32,33,36], and South Africa [46] reported that some parents were satisfied with the treatment and communication of vaccination information they received from the healthcare workers, and they were more accepting of routine childhood vaccination. In Cameroon, some parents indicated that they were happy because the healthcare facility was clean and the healthcare workers received them very well, 

*“I really like it. I like that it is clean and they are welcoming. I never have any worries”*. (Cameroon, participant quote) [30]

Other parents in Ethiopia mentioned that healthcare workers were also helpful [36]. Some parents in South Africa mentioned that they were very happy with the healthcare workers vaccinating their children because they were professional, friendly, and had positive attitudes. These parents added that the workers are patient, they don’t shout at them, and they answer any question and concern they have [46]. Some parents in Chad also revealed that healthcare workers were completely open to their questions about vaccination [29]. A study in Ethiopia indicated that some parents received information about vaccines from the healthcare workers before their children received vaccines [32]. Similar reports were made by a study conducted in Botswana, where some parents mentioned that healthcare workers provided them with information that made them to accept vaccination of their children [26].

Finally, one study conducted in Ethiopia revealed that some parents who were mistreated and humiliated by healthcare workers during the vaccination process for their children thought it was for a good cause to be treated as such: 

*“it is for our own good that they treated us in such a way”*.(Ethiopia, participant quote) [35]

## 4. Discussion

This systematic review focused on the factors that influence caregivers’ views and practices regarding routine childhood vaccination. Our findings were categorised into five main themes. Firstly, parents may be influenced by their ideas and practices surrounding health and illness. For example, parents may be vaccine hesitant due to their religious beliefs. Parents may also be more or less accepting of vaccination due to their beliefs around the benefits and risks of vaccines. Secondly, parents’ views and practices may be influenced by the views and practices of the social networks in which they reside. That is, parents’ vaccine hesitancy or acceptance of childhood vaccination may be influenced by community members including other parents, relatives, peers, neighbours, and other important members of their social network. Thirdly, wider political events and relations may also influence parents’ views and practices around childhood vaccination. Parents may be vaccine hesitant due to their distrust of those involved with vaccination programmes, such as the government or vaccine manufacturing companies. Fourthly, parents may be influenced by the level of their knowledge around childhood vaccination. Some parents were vaccine hesitant due to the lack of adequate information or understanding of the benefits of vaccination or the process of getting their children vaccinated. Finally, parents’ vaccination ideas and practices may be influenced by their access to, and experiences of, vaccination services. For example, parents may be vaccine hesitant due to socioeconomic challenges such as distance to healthcare facilities, lack of finances, household work, employment, or childcare constraints. Parents may also vaccine hesitant due to the undesirable features of vaccination services and delivery logistics such as long waiting times at healthcare centres, vaccine stockouts, or the requirement to present the vaccination booklet before vaccination. Parents’ acceptance or hesitancy to childhood vaccination may also be influenced by their interactions with healthcare workers; whether they were well treated or mistreated. 

Our systematic review was comprehensive. We included studies from different countries and types of settings in Africa. Therefore, our findings may be applicable to all of Africa, as well as other settings that are similar to those where the studies were conducted. All studies were conducted among caregivers, defined as parents (mothers or fathers) or other people with parental responsibilities, except for one study which targeted pregnant women. Most of the included studies focused on routine childhood vaccines. 

The search for eligible studies covered the period from 1974 to 10 July 2020. Therefore, we might have missed findings from studies that were published after 10 July 2020. Therefore, findings from these studies are needed. 

Our findings from this review have both commonalities and dissimilarities with findings from the Cochrane review by Cooper and colleagues [5]. The Cochrane review focused on the factors that influence caregivers’ views and practice regarding routine childhood vaccination worldwide, while the current review is specifically focused on Africa. The 27 studies that are included in the current review were part of those studies that met the inclusion criteria in the Cochrane review, but 88% of them were not sampled for analysis. Of the 27 studies included in the current review, 6 of them were part of the sampled studies in the Cochrane review. 

Unlike the Cochrane review, our findings are categorised into five analytical themes, while those of the Cochrane review were divided into four themes. The fifth main theme identified in the current review, but not in the Cochrane review, is lack of knowledge or information (Theme 5). In this theme our review found that lack of knowledge or information on childhood vaccination can influence parents’ rejection of vaccination of their children. These parents complained that their children were vaccinated without being told any information about vaccines and diseases that they targeted. Our findings were not surprising at all because the review was focusing on Africa, a continent compromising of many low and middle-income countries facing many challenges such as limited human resources and low level of health literacy. 

Our findings were also categorised into 9 subthemes compared to 15 in the Cochrane review. The six sub-themes of the Cochrane review that our findings were not categorised into were under Theme 1 (The ‘fragile’ infant, Primacy of ‘nature’ and ‘the natural’; Individualised health, immunity, and vaccine-response trajectories; Claiming parental expertise; and Personal choice and responsibility) and Theme 2 (Vaccination ideas and practices shape social networks). The Cochrane review found that some parents were more and less accepting of childhood vaccination due to the fragility of their infants towards diseases and effects of vaccination. It also found that some parents were vaccine hesitant because of understanding of health and illness as wholistic or naturalistic. Cooper and colleagues also noted in the Cochrane review that many parents, when deciding whether or not to have their children vaccinated, consider that children have unique bodies and immune systems and therefore have individual vaccine needs and susceptibilities. In contrast to our findings, the Cochrane review also found that many parents had a view of themselves as experts of their children, having the best understanding of their children’s unique health strengths and vulnerabilities. The Cochrane review also found that many parents held a perception that healthcare-related decision-making, including vaccination, are matters of personal responsibilities and choices. Finally, the Cochrane review also found that many parent’s ideas and practices around childhood vaccination also influenced their social networks. That is, the vaccination ideas and practices they shared were a strong factor in building social relationships and bonds, and associated access to social support and resources. These differences confirm the context specificity of vaccine acceptance and hesitancy.

Similar to the Cochrane review, the four main themes that Cooper and colleagues divided their findings into were also identified by us in the current review, namely, Ideas and practices surrounding (child) health and illness (Theme 1); Social communities and networks (Theme 2); Political events, relations, and processes (Theme 3); and Access-supply-demand interactions (Theme 4). As with the current review, the Cochrane review found that some parents were less willing to vaccinate children because of their religious beliefs. Like the findings of Cochrane review, we also found that parents’ views and practices regarding childhood vaccination were influenced by the vaccination views and practices of the social networks in which they live. However, our review found that fathers were the main decision-makers for vaccinating their children, whereas mothers were in the Cochrane review. Similar to our findings, the Cochrane review found that some parents were hesitant to vaccinate their children because of their distrust of institutions or systems associated with vaccination. Other similar finding were that some parents’ vaccination views and practices regarding childhood vaccination were shaped by their access to and experiences with the facilities where vaccinations are administered and by frontline healthcare workers. This was the most represented theme in our findings when compared to other themes our findings were categorised into. As with the Cochrane review, we found that many parents were vaccine hesitant because of several socioeconomic challenges they faced when trying to access vaccination services, including distance, transportation, household work, and childcare and work constraints. Similar to our review, the Cochrane review also found that some parents were less accepting of childhood vaccination because of undesirable features of vaccination services and delivery logistics, such as facility waiting times, vaccine stock outs, and constraining organisational procedures. As with the findings of the Cochrane review, we also found that parents’ decisions to vaccinate their children or not was influenced by the manner in which they were treated by the healthcare workers. 

The questions below, derived from our findings, may help programme managers, policy makers, and other decision makers when planning and implementing strategies to promote childhood vaccination acceptance and uptake. Since our findings are both similar and different to the findings of the Cochrane review by Cooper and colleagues, these questions are also similar and different to various implications for practice identified in the Cochrane review. 

As with the Cochrane review, we propose that decision makers consider the following questions when planning and implementing interventions for reducing vaccine hesitancy and increasing vaccine acceptance and uptake: Have you considered how the intervention(s) could be tailored to the specific health beliefs and practices of parents in your target setting, for example, through immunization communication that acknowledges, aligns with, and builds on parents’ specific health beliefs and practices?Have you considered whether the intervention(s) might involve the social groups to which the parents in your target setting belong, e.g., by involving influential people within those groups (e.g., key opinion leaders) in the design, planning, and/or implementation of the intervention(s)?Have you considered whether the intervention(s) could be tailored to the specific reasons for parents’ mistrust, such as dialogue-based approaches that invite open discussion about the reasons for mistrust, or providing a broader range of essential services or commodities along with vaccination? Alternatively, have you considered working with groups or individuals known by parents to be trusted sources of information (e.g., nongovernmental organisations, local opinion leaders, etc.) and possibly involving them in the design, planning, or implementation of the intervention(s)?Have you considered whether the intervention(s) could target the specific barriers parents face in accessing immunizations, such as providing outreach immunizations or mobile immunisation teams that bring immunizations closer to parents’ homes?Have you considered whether the intervention(s) could be tailored to address specific characteristics of immunisation services that might negatively impact parent acceptance of immunisation in your target setting? For example, if your health facility regularly experiences shortages of vaccine supplies, could you identify what causes these shortages and how these problems might be addressed? Could the logistical procedures that parents must follow for vaccination possibly be redesigned to better meet parents’ needs and circumstances? Could the physical environment at your health facility be redesigned to allow for more efficient delivery of immunizations?Have you considered intervention(s) that specifically target healthcare workers, for instance by making them aware of the influence their interactions with parents may have, providing them with training in communication skills or increased supervision and support, or adapting the kinds of vaccination information healthcare workers have access to and provide to parents. Unlike the Cochrane review, we found that many parents were less accepting of childhood vaccination due to the lack of knowledge or information around vaccines. Therefore, based on this finding of our review, the question that may help when planning and implementing strategies to promote childhood vaccination acceptance and uptake is as follows:Have you considered intervention(s) that specifically target parents’ lack of information or knowledge regarding childhood vaccination in your setting, for instance, by providing vaccine information that informs and educates parents about the benefits of vaccines and the process of vaccination?

## 5. Conclusions

Our review found that parents’ views and practices regarding childhood vaccination in Africa are influenced by various factors. As hypothesised, our findings were both similar and different to those of the Cochrane review published in 2021 by Cooper and colleagues. All of the themes identified in the Cochrane review were also identified in our review. This strengthens both the validity and applicability of the four-theme framework generated by the Cochrane for understanding the factors influencing parent’ views and practices regarding childhood vaccination in Africa. This current review identified one theme (‘lack of information or knowledge’) that was not identified in the Cochrane review. Comprehensively understanding the influencing factors in Africa may therefore necessitates considering issues related to knowledge and information. This will also help programme managers, policy and decision makers, and other stake holders in Africa, to promote acceptance and uptake of childhood vaccination by developing and implementing interventions that are tailored to address lack of knowledge and information around vaccines. 

## Data Availability

Not applicable.

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
