# Peer review of "A Systematic Review of Factors That Influence Parents’ Views and Practices around Routine Childhood Vaccination in Africa: A Qualitative Evidence Synthesis"

_vaccines, 2023, doi:10.3390/vaccines11030563_

Round 1
Reviewer 1 Report
The title should reflect that you focus on parents, and perhaps that this is a study with a rather qualitative approach ?
The rationale for this additional analysis is very well explained in the introduction, but I would recommend to make this also somewhat clearer in the abstract, and along the same line if the 27 articles generates ‘other information’ compared to the 6 initially included. After reading the full version, it seems that you have tried to extract ‘qualitative’ information, and that’s fine, but perhaps this should also be somewhat better reflected in the abstract.
Perhaps a summarizing table at the beginning of the discussion could be useful to make the paper more accessible to the readership ?
Author Response
Responses to the reviewer
|
Comment |
Response |
|
The title should reflect that you focus on parents, and perhaps that this is a study with a rather qualitative approach?
|
Thanks for the comment. The title has been revised. We have included parents. See line 2. |
|
The rationale for this additional analysis is very well explained in the introduction, but I would recommend to make this also somewhat clearer in the abstract, and along the same line if the 27 articles generates ‘other information’ compared to the 6 initially included. After reading the full version, it seems that you have tried to extract ‘qualitative’ information, and that’s fine, but perhaps this should also be somewhat better reflected in the abstract. |
The rational of additional analysis is now explained in the abstract. See line 14 – 33. |
|
Perhaps a summarizing table at the beginning of the discussion could be useful to make the paper more accessible to the readership?
|
A table summarizing our findings is now included. See line 940. We have also added a table about the characteristics of the studies included in this review. See line 918 |
Reviewer 2 Report
Dear authors
It was with great pleasure that I reviewed your manuscript.
The systematic review is well done.
However, it would be more pleasant to read if some of the results of the studies were presented in tables.
Therefore, I would suggest you put tables with the summary of the studies analyzed.
If they say that many parents avoid the vaccine, it is different from putting in a table the number of parents who avoid vaccinating their children. The way they did it was a bit vague.
My Best Regards
Author Response
Response to reviewer
|
Comment |
Response |
|
It was with great pleasure that I reviewed your manuscript. The systematic review is well done. However, it would be more pleasant to read if some of the results of the studies were presented in tables. Therefore, I would suggest you put tables with the summary of the studies analyzed. If they say that many parents avoid the vaccine, it is different from putting in a table the number of parents who avoid vaccinating their children. The way they did it was a bit vague. |
Thanks for the comment. A table summarizing our findings is now included. See line 924. We have also added a table about the characteristics of the studies included in this review. See line 918 |
|
|
|
|
|
|
Reviewer 3 Report
Thank you for the invitation to review this manuscript. This manuscript provides a summary of 27 studies conducted in African regions on vaccine hesitancy. The authors have focused on the factors associated with VH. I have concerns about the methodology of this review. The authors have conducted a previous Cochrane review and extracted 6 studies from a previous review. The conclusion and results of these 6 studies have been described in reference no. 5. In this context, what is the rationale of the current review? I have gone through the previous review where the VH is discussed in a detailed way in Africa. Cochrane reviews are aimed to provide a comprehensive review to generate a final conclusion for practice and policy. I am not sure why other studies were not included in this review. I believe that there are more than 6 studies from the African regions. The intention of the authors is to underscore the phenomena of VH in Africa, but my concern is that only six studies can generalize the whole continent? I believe that inclusion of more studies, and stratification of results in a systematic review manner will provide more useful information and will add novel points to the existing literature. The factors highlighted in this study have already been discussed in reference no. 5.
The authors have claimed that only one factor "lack of information or knowledge" is identified through this review while all other factors have been discussed previously. I am not convinced with the rationale, novelty, and generalizability of this review as the factors discussed in the current study are well-known and established in literature.
Author Response
Response to reviewer
|
Comment |
Response |
|
Thank you for the invitation to review this manuscript. This manuscript provides a summary of 27 studies conducted in African regions on vaccine hesitancy. The authors have focused on the factors associated with VH. I have concerns about the methodology of this review. The authors have conducted a previous Cochrane review and extracted 6 studies from a previous review. The conclusion and results of these 6 studies have been described in reference no. 5. In this context, what is the rationale of the current review? I have gone through the previous review where the VH is discussed in a detailed way in Africa. Cochrane reviews are aimed to provide a comprehensive review to generate a final conclusion for practice and policy. I am not sure why other studies were not included in this review. I believe that there are more than 6 studies from the African regions. The intention of the authors is to underscore the phenomena of VH in Africa, but my concern is that only six studies can generalize the whole continent? I believe that inclusion of more studies, and stratification of results in a systematic review manner will provide more useful information and will add novel points to the existing literature. The factors highlighted in this study have already been discussed in reference no. 5.
The authors have claimed that only one factor "lack of information or knowledge" is identified through this review while all other factors have been discussed previously. I am not convinced with the rationale, novelty, and generalizability of this review as the factors discussed in the current study are well-known and established in literature.
|
Many thanks for this feedback and thorough engagement with our review. We fully agree that Cochrane reviews are aimed to provide a comprehensive review to generate a final conclusion for practice and policy. However, when it comes to Cochrane reviews of qualitative evidence, standard methodological practice involves sampling of studies to be included in the analysis (See Cochrane requirements for reviews of qualitative evidence: https://epoc.cochrane.org/news/qualitative-evidence-synthesis-template). This is because these reviews aim to achieve conceptual, rather than, statistical generalizability. For the original review we also employed a meta-ethnographic synthesis approach which necessitates that only a few conceptually rich studies (including books- which we included 3) be included so that new concepts or theory might be generated (See standard guidance for meta-ethnographies: https://bmcmedresmethodol.biomedcentral.com/articles/10.1186/s12874-018-0600-0). These are key methodological and epistemological differences between systematic reviews of effectiveness and those of qualitative evidence which we hope can be appreciated.
For this current review, we decided to relook at all the studies identified within the African region to determine whether inclusion of these additional studies changed the concepts and theory developed from the larger review. Again, this is a common practice in the realm of Qualitative Evidence Synthesis, particularly a "best fit" framework synthesis approach which specifically seeks to test existing qualitative findings and conceptual frameworks (See eg: https://bmcmedresmethodol.biomedcentral.com/articles/10.1186/s12874-019-0665-4 AND https://bmcmedresmethodol.biomedcentral.com/articles/10.1186/1471-2288-11-29). It is an approach that is used to enhance the generalizability of these findings/frameworks, which is where we feel the strength of this current review lies. That is, we have shown the validity and applicability of the 4-theme framework for understanding the factors influencing parent’ views and practices regarding childhood vaccination in Africa. We have also shown how comprehensively understanding these factors in our region also necessitates considering adding an 5th theme, not identified in the original Cochrane review. Ultimately, we believe these insights are important contributions to the evidence-base, theoretically and methodologically.
We have revised our description of the aims and focus of this current review, so its rationale, novelty, and generalizability are clearer (See line 67 - 101
We have also added some details in the conclusion which more clearly highlights how the findings of this review make an important contribution to the evidence-base (See line 888 - 904)
|
Round 2
Reviewer 2 Report
Dear authors
Thank you for making the changes I proposed.
My best regards
Reviewer 3 Report
Thank you for addressing my concerns.